# TextEconomizer: Enhancing Lossy Text Compression with Denoising Autoencoder and Entropy Coding

## Abstract

Lossy text compression reduces data size while preserving core meaning, making it ideal for summarization, automated analysis, and digital archives where exact fidelity is less critical. While extensively used in image compression, text compression techniques, such as integrating entropy coding with autoencoder latent representations in Seq2Seq text generation, have been underexplored. A key challenge is incorporating lossless entropy coding into denoising autoencoders to improve storage efficiency while maintaining high-quality outputs, even with noisy text. Prior studies have mainly focused on near-lossless token generation with little attention to space efficiency. In this paper, we present a denoising autoencoder with a rectified latent representation that compresses variable-sized inputs into a fixed-size latent space without prior knowledge of dataset dimensions. By leveraging entropy coding, our model achieves state-of-the-art compression ratios alongside competitive text quality, as measured by diverse metrics. Its parameter count is approximately 196 times smaller than comparable models. Additionally, it achieves a compression ratio of $67\times$ while maintaining high BLEU and ROUGE scores. This significantly outperforms existing transformer-based models in memory efficiency, marking a breakthrough in balancing lossless compression with optimal space optimization.

## 1 Introduction

Text compression is an essential part of data compression that involves diminishing the volume of textual data while preserving its informational content. This process is divided into lossless compression, which allows full data recovery, and lossy compression, which strategically sacrifices details to achieve higher compression ratios. The exponential growth of digital information has introduced noteworthy challenges in storage and transmission efficiency (Office, 2023), specifically in contexts where exact textual reproduction is not compulsory. Lossy text compression has numerous applications; lossy text storage and compression offer practical solutions for non-critical documents and archiving, where exact preservation of every detail is not essential. This approach is useful for internal reports, outdated or old document versions, corporate archives, and educational institutions, where key information needs to be retained, but minor errors or formatting loss won't impact usability. Libraries, digital archives, and universities can compress vast collections, such as research papers or public domain books, to save storage space while maintaining accessibility. Additionally, chatbots, email archiving, and web search engines benefit from lossy text processing, enabling fast retrieval and facilitated communication. This makes lossy text compression a valuable, efficient tool for handling large-scale text data while optimizing storage and retrieval performance.

Image compression harnessing neural networks, especially through Variational Autoencoders, has gained prominence (Geleta et al., 2023), meanwhile myriad approaches have been discovered to reduce textual volume while preserving salient information. Existing research underscores the efficacy of transformer-based models, including BERT (Li et al., 2023), LLaMA (Valmeekam et al., 2023), and ALBERT (Li et al., 2021), LSTM (Prato et al., 2019), in maintaining contextual integrity during decompression across diverse linguistic landscapes. In particular, architectural modifications such as the shared encoder have shown strong performance (Li et al., 2020). Furthermore, chasing the trend of fine-tuning domain-specific pre-trained models and incorporating the LoRA (Hu et al.,

2021) technique has also resulted in praiseworthy results (Ge et al., 2023). Cross-lingual augmentation strategies enhance the capabilities of transformer models for languages with diverse resource availability, an aspect that has not been investigated in the work of (Mao et al., 2022). Notwithstanding these advancements, recent studies have encountered limitations. The study conducted by Huang et al. (Huang et al., 2023) leveraged the operational procedure of arithmetic coding and incorporated it with GPT for lossless text compression, lacking a quest for harnessing general-purpose compressors for an extensive comparison of compression ratios. Additionally, it is important to note that non-autoregressive decoding may not flawlessly recover the original text, and iterative inspection of meaning preservation might be time-consuming (Ge et al., 2022). (Ge et al., 2023) employed a LoRA-configured Llama-2-7b for context compression, albeit with augmented parameters, exacerbating the already prodigious parameter count characteristic of contemporary LLMs. Meanwhile, (Qin et al., 2023) executed an autoencoding task generating dynamic text segments with residual connections, achieving a mere tenfold compression—a ratio deemed insufficient. Furthermore, (Wang et al., 2021) provided an autoencoding model trained to reconstruct input texts by means of a combination of token embeddings as a bottleneck, a strategy susceptible to overfitting, while the compression ratio $r$ is lower. Strengthening the fixed-size bottleneck remains a pivotal challenge for Transformer-based large language models (LLMs) due to their intrinsic self-attention mechanism. While prior research ((Rae et al., 2019; Malireddy et al., 2020; Wang et al., 2021)) has explored text compression in LLMs, they frequently grapple with the challenge of mitigating memory intricacy. Despite their capacity for fixed-size latent spaces, LSTM-based autoencoders frequently produce inadequate results in autoregressive decoding tasks. Using these orthodox approaches, we tackle the issue of memory complexity from an alternative standpoint: employing lossy text compression.

In this study, we introduce TextEconomizer, a uniquely tailored autoencoder-based approach designed for English autoencoding tasks, guided by a sophisticated noisy text process, consisting of a single encoder-decoder layer built upon a bidirectional gated recurrent unit (GRU). TextEconomizer optimizes the balance between model complexity and computational efficiency, while simultaneously improving space efficacy that enhances qualitative text performance through attention mechanisms and boosts quantitative compression efficiency by continuously refining the fixed-size bottleneck. Besides, the integration of entropy coding further compresses the latent representation, solidifying TextEconomizer as a memory-efficient monolingual method. The contribution of our study is summarized below:

- A pragmatic text noise process is tailored to encompass a wide range of text distortions, allowing the neural network to uncover and learn to fix multifarious mistakes during training.
- A monolingual autoencoder-based method called TextEconomizer has paved the way for enhanced performance in autoencoding tasks, extending the relevance of autoencoding beyond image compression to text-based applications.
- Benchmarking across diverse corpora demonstrated that TextEconomizer acquires state-of-the-art memory efficiency with negligible degradation in text quality.
- We rigorously evaluated the efficiency of the refined latent representation in capturing intricate linguistic patterns, further compressing it by harnessing entropy coding.
- An exploration into the impact of the size of the training corpus on the efficacy of TextEconomizer in restoring identical text within English text is conducted, shedding light on its potential contributions in the realm of decompression.

## 2 RELATED WORK

The domain of text compression has become widespread attention of research, attracting significant interest and contributing to the development of innovative methodologies and datasets. Our extensive study aims to summarize key findings and showcase the evolving landscape of text compression methodologies across different language contexts, including lossy (Li et al., 2023; 2020; 2021; Ge et al., 2022) and lossless (Valmeekam et al., 2023; Huang et al., 2023; Mao et al., 2022) techniques.

The advent of text compression has seen the employment of transformer-based methods where GPT, Llama, BART, and a single-layer transformer have been used. Among lossy techniques, (Li et al.,

2023) introduced an innovative method to compress English text by masking less important words and then restoring them using a Transformer-based model. Among compressive-memory-based methods, (Rae et al., 2019) introduced a refined extension of the Transformer architecture (Vaswani et al., 2023), employing a compressive-memory-based approach that shrinks past activations into more consolidated representations instead of discarding them.

Encompassing Transformer-based lossy autoencoding and translation, (Ge et al., 2023) proposed an innovative approach by leveraging the capabilities of the Llama model (Touvron et al., 2023) to generate pertinent memory slots incorporating the teacher-forcing mechanism for both autoencoding and machine translation tasks. Whereas (Qin et al., 2023) have introduced an interesting methodology that uses the BART (Lewis et al., 2019) encoder for producing highly significant dynamic text segments, called "NUGGETS", by distilling the logits through a feed-forward network.

Within LSTM-based autoencoding tasks, (Malireddy et al., 2020) introduced an indicator vector to signify the presence or omission of each word and eliminate less pertinent words. Notwithstanding the aforementioned approach, (Tissier et al., 2019) introduced an autoencoder-based model that condenses real-valued embedding vectors into fixed-length binary representations. Contrariwise, (Acharya et al., 2019) used a similar approach, decomposing and transforming the embedding layer through matrix factorization, and employing lower-rank matrices to enhance storage efficiency.

Our study has also revealed that lossless text compression techniques have yielded exceptional results by incorporating diverse techniques with transformers. (Valmeekam et al., 2023) proposes a novel method that uses the LLM to predict the next token (based on probability ranking) in a text sequence based on a window of past tokens. (Huang et al., 2023) introduced a novel method that utilizes the GPT model to calculate probability distributions for each token and represent the entire text with a single number. Additionally, (Delétang et al., 2023) compared predictive models and lossless compressors and recommended using large self-supervised language models for compression.

Transformer-based techniques, though superior in autoregressive decoding evaluation, encounter memory constraints. Despite these successes, the application of latent representations for text compression remains underexplored. It is noteworthy that transformers require prodigious datasets for optimal performance.

# 3 CORPUS CREATION

## 3.1 DATA SOURCING

The source of our data is from four publicly available standard large-scale corpora: WMT19, PwC, WMT14, and BookCorpus. The WMT19 corpus encompasses 26 million ZH-EN language pairs, while PwC contains 242k samples structured as (input, prompt, and answer) triads. WMT14 comprises 1.6 million EN-FR sentence pairs and the BookCorpus dataset contains 7.8 million English sentences.

## 3.2 DATA PREPROCESSING

From the WMT19 corpus, we extracted the English sentences from the Chinese-to-English pairs. We also obtained English text from WMT14's English-to-French pairs. For the PwC dataset, we isolated the answer column. Since BookCorpus is monolingual, no extraction was needed. This systematic approach produced an English-centric corpus across all datasets. In our text preprocessing, we delineate a comprehensive character set containing 80 frequently occurring English characters, denoted as $DC = \{DC_1, DC_2, \ldots, DC_{80}\}$. This set is augmented by 14 frequently occurring punctuation marks in English, represented as $PM = \{PM_1, PM_2, \ldots, PM_{14}\}$, and a space character $SP$. The resulting amalgamated set of 95 English characters is defined as $C = \{DC + PM + SP\} = \{C_1, C_2, \ldots, C_{95}\}$. Subsequently, we consider each sentence in our corpus, denoted as $S = \{S_1, S_2, \ldots, S_N\}$, where $N$ illustrates the total number of characters in the sentence. We then employ an iterative technique, examining each character $S_i \in S$ and systematically eradicating any character not present in our predefined character set $C$. This meticulous preprocessing assures a standardized and sophisticated textual dataset for subsequent neural network processing.

## 3.3 DATA AUGMENTATION

We introduce a sophisticated noise injection technique to introduce controlled linguistic variability. This process operates on the premise that each sentence constitutes a finite set of lexical units, denoted as $S = \{W_1, W_2, \ldots, W_{N-1}, W_N\}$ where $N$ is the length of the sentence such that $N \in Z^+$. The proposed noisy text corruption process aims to forge an altered version of the input sentence S that preserves its semantic meaning while incorporating realistic linguistic perturbations. The process commences with identifying named entities $NE = \{e_1, e_2, \ldots, e_m\}$ utilizing a Named Entity Recognition (NER) model. Subsequently, the sentence undergoes identifying part-of-speech (POS) tags $T = \{(W_i, t_i), (W_2, t_2), \ldots, (W_N, t_N)\}$ are assigned. Auxiliary verbs $V_a \subseteq S$ are probabilistically omitted with a probability $P_{aux}$, constrained to words where $W_i \in V_a$ and $t_i \in V^*$, where $V^*$ encompasses all verb forms. Consequently, we obtain the modified set $\tilde{S} = \{\tilde{W}_1, \tilde{W}_2, \ldots, \tilde{W}_{M-1}, \tilde{W}_M\}$. The corruption process is controlled by a normally distributed corruption probability $p_c \sim \mathcal{N}(\mu_p = 0.6, \sigma_p^2(\sigma = 0.1))$, which is bounded by a maximum corruption threshold $p_{max} = 0.5$ to ensure the degree of alteration remains within acceptable limits. Given $p_c$, we determine the number of words to be corrupted as $k = \lceil M \times p_c \rceil$. These words are carefully chosen to avoid consecutive corruptions, preserving the sentence structure. For each chosen word $\tilde{W}_i$, the corruption method is selected based on POS tags and whether $\tilde{W}_i \in NE$ or critical nouns and verbs $C = \{c_1, c_2, \ldots, c_l\} \subset \tilde{S}$. Specifically, if $\tilde{W}_i \notin NE$ and $t_i \in \{P_{nouns}, P_{verbs}, P_{adj}\}$, a contextual synonym $W_i^{'}$ is generated using a masked language model with a $p_{mlm} = 0.5$ probability, unless $W_i^{'} = \tilde{W}_i$. Alternatively, spelling augmentation is propagated $p_{spelling} = 0.3$ of the time, while random word substitution is employed in the remaining $p_{sub} = 0.2$ of cases. Words that do not fall into the categories of nouns, verbs, or adjectives are subjected to typographical alterations through character interchanges or replacements. Punctuation corruption is then applied with a $p_{punc} = 0.2$ probability per word, eradicating existing punctuation or introducing new punctuation marks. This meticulous process pinnacles in a corrupted sentence $S^{'}$ that mimics natural language errors, echoing common mistakes while preserving the essential meaning of the original sentence $S$.

## 3.4 CORPUS STATISTICS

In our curated English corpus, characterized by intended noise injection, we have selected $\approx$ 246K, 600K, 600K, and 1M source-target pairs from the PwC, WMT14, WMT19, and BookCorpus datasets, respectively, due to resource limitations. Within these pairs, the source sentences undergo a meticulous text noise technique, while the target sentences serve as pristine, noise-free counterparts. To achieve this, we systematically introduced perturbations into each sentence, as outlined in Subsection 3.3. Furthermore, the corpus exhibits the following linguistic statistics: PwC demonstrates a minimum of 1, a maximum of 180, and a mean of 35.52 words per sentence; WMT14 ranges from 1 to 72 words, averaging 21.68; WMT19 spans from 1 to 137 words, with a mean of 12.67; and BookCorpus holds sentences varying from 2 to 150 words, averaging 15.13 words per sentence.

# 4 METHODOLOGY

## 4.1 PROBLEM FORMULATION & OVERVIEW

Consider input sentence $S = \{S_1, S_2, \ldots, S_{N-1}, S_N\}$, where $N$ denotes the word count. The noise injection process ($\aleph(\cdot)$) adroitly introduces strategic realistic noise through a meticulous automated supervision protocol. Subsequently, we scrutinize the juxtaposition of token pairs, $X_I = \{x_1, x_2, \ldots, x_{n-1}, x_n\}$ and $Y_I = \{y_1, y_2, \ldots, y_{k-1}, y_k\}$, where $X_I$ epitomizes the input sequence with multifarious noise types, while $Y_I$ embodies the pristine target sequence. The corrupted input sequence $X_I$ undergoes tokenization through a pre-trained tokenizer $T(\cdot)$, before being fed to the encoder $E(\cdot)$, which engenders context vectors of the sentence, denoted as $\mathcal{Z}_{512}$. This latent representation $\mathcal{Z}$ is further subjected to compaction through the Lempel–Ziv–Markov compressor $\mathcal{LC}(\cdot)$ to facilitate parsimonious storage on a hard disk, whereupon memory-related computations are conducted. The Lempel–Ziv–Markov decompressor $\mathcal{LD}(\cdot)$ reconstitutes the compressed representation to a form indistinguishable from $\mathcal{Z}$. The decoder $D(\cdot)$ uses this latent representation along with previously generated tokens to remove the noise and autoregressively synthesize the correct sentence. This preprocessing pipeline exhibits seamless integration capabilities with any RNN and

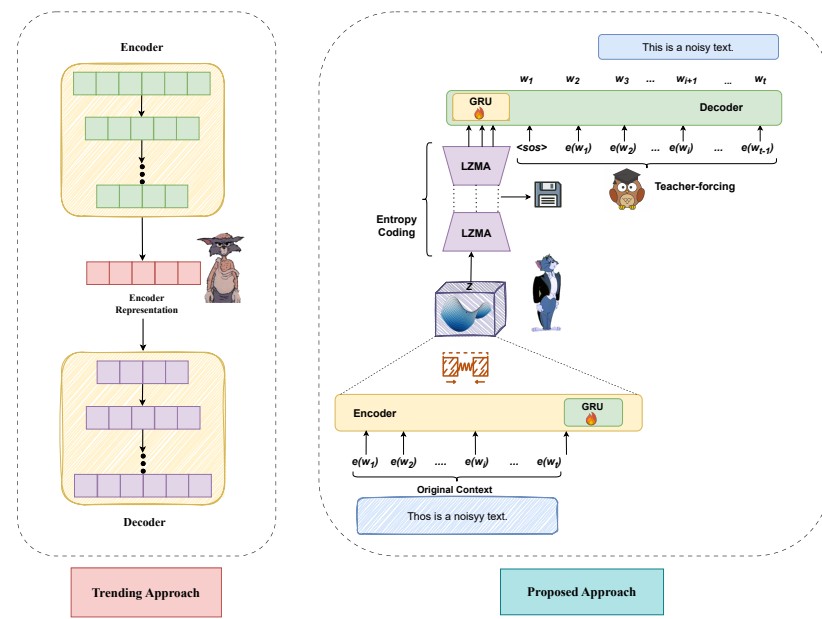

Figure 1: **(Left)** Trending Transformer-based approach that generates a latent representation with dimensions identical to the input. **(Right)** TextEconomizer employs a fixed-size latent representation.

Transformer-based encoder-decoder architecture. The entire procedure can be encapsulated in the following mathematical formulation:

$$\hat{Y} = D((\mathcal{LD}(\mathcal{LC}(E(T([X_I]), W^E))), D_{out}^{t-1}), W^D) \quad (1)$$

## 4.2 TEXTECONOMIZER

In this section, we delve into the details of TextEconomizer.

### 4.2.1 ENCODER

Given an input sequence of tokens $\mathbf{X} = \{x_1, x_2, \ldots, x_n\}$, where n denotes the sequence length, we assigned unique discrete values to each lexical unit. To ensure uniform input dimensionality, we augmented individual input sequence $\mathbf{X_i}$ by incorporating padding. Subsequently, each token $x_i$ went through an embedding layer $\mathbf{E}$, converting discrete values into continuous vectors, represented by the trainable matrix $\mathbf{E}_{x_i} = \mathbf{Embedding}(x_i)$. To mitigate overfitting, we applied dropout to the embeddings, yielding $\mathbf{DE}_{x_i} = \mathbf{Dropout}(\mathbf{E}_{x_i})$. Through backpropagation during training, these metrics are iteratively refined to minimize the loss function. The resultant $\mathbf{DE}_{x_i}$ is then propagated through $\mathbf{K}$ bi-directional Gated Recurrent Unit (GRU) layers. These layers, predicated on four primary components—update gate $z_t$, reset gate $r_t$, candidate hidden state $\tilde{h}_t$, and final hidden state $h_t$—process the input bidirectionally. This bidirectional GRU architecture produces hidden states from both directions, which are subsequently concatenated, yielding the output:

$$\mathbf{O}_{enc}, \mathbf{h}_t = \mathbf{BidirectionalGRU}(\mathbf{DE}_{x_i}) \quad (2)$$

In the ensuing step, the hidden states of the forward GRU ($\mathbf{h}_{\text{fwd}}$) and the backward GRU ($\mathbf{h}_{\text{bwd}}$) undergo concatenation before feeding into a feed-forward layer. This feed-forward mechanism introduces non-linearity to the encoder through a linear transformation incorporating the hyperbolic tangent function ($\tanh$). This non-linear activation function aids in enhancing the obtained representation, which constitutes the initial decoder hidden state:

$$\mathbf{Z} = \tanh(\mathbf{FFN}(\mathbf{h}_{\text{fwd}} \oplus \mathbf{h}_{\text{bwd}}) \quad (3)$$

This resultant $\mathbf{Z}$ represents the latent space—a compressed, lower-dimensional input representation. The outputs of this sophisticated encoder architecture serve as contextual representations of the

input sequence $\mathbf{X}$ covering local and global dependencies, primed for harnessing by the attention mechanism in the subsequent decoding phase.

### 4.2.2 DECODER

The target sequence denoted as $\mathbf{Y} = \{y_1, y_2, \ldots, y_m\}$, where $m$ represents the sequence length, initially transformed through an embedding layer, such that $\mathbf{E}y_i = \mathbf{Embed}(y_i)$, transmuting discrete token indices into continuous vector representations. To mitigate overfitting, a dropout mechanism is applied to the embedded tokens, yielding $\mathbf{DE}y_i = \text{Dropout}(\mathbf{E}_{y_i})$. At each decoding step $t$, the model leverages the hidden state from the antecedent time step, $\mathbf{H}_{t-1}$, incorporating the encoder outputs to compute attention weights. The attention mechanism (Bahdanau, 2014), a cornerstone of our model's architecture, uses the latent representation $\mathbf{Z}$ and the encoder outputs, $O_{enc_t}$, where $n$ denotes the length of the source sequence, to calculate attention scores. These scores, $\alpha_t$, quantify the relevance of each encoder output to the current decoding step. The scores undergo normalization through softmax, pinnacling in the computation of a context vector through a weighted summation of the encoder outputs. This process is depicted in the following mathematical formulation:

$$\mathbf{C}_t = \sum_{i=1}^{n} \text{softmax}\left(\mathbf{V}^T \cdot \tanh\left(\mathbf{W}_1 \cdot [\mathbf{Z}; \mathbf{h}_i^{enc}]\right)\right) \cdot \mathbf{h}_i^{enc} \tag{4}$$

The resultant context vector $\mathbf{C}_t$ is concatenated with the current embedded target token $\mathbf{DE}y_t$ to form the input for the Gated Recurrent Unit (GRU) at time step $t$. This concatenated input propagated through $K$ GRU layers, which update the hidden state based on the current input and the latent representation $\mathbf{Z}$. Consequently, the output $\mathbf{O}_{dec_t}$ of the GRU and the hidden state $\mathbf{H}_t$ are updated. Subsequently, a threefold concatenation of the output $\mathbf{O}_{dec_t}$, the context vector $\mathbf{C}_t$, and the embedded target token $\mathbf{DE}y_t$ is passed through a feed-forward network (FFN). This operation introduces non-linearity to the model and pinnacles in the generation of a prediction for the succeeding token:

$$\hat{y}_t = \text{FFN}\left(\mathbf{O}_{dec_t} \oplus \mathbf{C}_t \oplus \mathbf{DE}y_t\right) \tag{5}$$

During the training phase, the model employs teacher forcing with a ratio of 0.5. This approach entails that at each time step $t$, with a probability of 50%, the model utilizes the actual target token $y_t$ as input for the subsequent time step. In the alternative scenario, the predicted token of the model $\hat{y}_t$ is used as input for the ensuing decoding step. This iterative process is performed for each token in the target sequence, culminating in refined target sequence representations, denoted as $\mathbf{Y}$ based on both the autoregressive projections and the encoder's contextualized representations.

### 4.2.3 HYPERPARAMETERS

The hidden dimension is kept as 512 through the encoder and decoder layers for maintaining consistency. To support the model's depth and capacity the number of neurons is kept at 1024 for the feed-forward layer of the encoder, facilitating consistency in the bottleneck representation, and a 0.5 dropout ratio is applied to prevent overfitting. To maintain efficient computation and non-linearity over the network, we have incorporated $\tanh$. The model underwent training until convergence incorporating cosine annealing to enhance gradual convergence incorporating AdamW optimizer with a minimum learning rate of $5 \times 10^{-5}$ starting from 0.001. We incorporated the cross-entropy loss having 0.1 label smoothing for the optimization process, which leads the model towards desired translations.

## 5 EXPERIMENTAL ANALYSIS

### 5.1 DATASETS

**WMT14 (Bojar et al., 2014)** For the WMT14 dataset we use the English-to-French (EN-FR) subset and the training, validation, and test sets comprise 600000 (600K), 3000 (3K), and 3003 (3k) source-target pairs respectively.

**WMT19 (Barrault et al., 2019)** In our experimental setup, we use the English-Chinese subset of the WMT19 corpus, selecting a sample of 600K source-target pairs for the training set and 3.98K pairs for the test set.

**Prompt-With-Context (PWC) (Ge et al., 2023)** The dataset encompasses ≈242K training instances and 18,100 test instances, subsequently altered by adding noise to the original sentences.

**BookCorpus (Zhu et al., 2015)** Following a meticulous text noise injection phase, we partitioned the corpus into 10,00,000 (1M) pairs in the training and 20,000 pairs in the test set to create distinct training and test sets.

## 5.2 BASELINES

**ICAE (Ge et al., 2023)** In-context autoencoder is a language model developed by Microsoft researchers, showcasing remarkable results for both autoencoding and language modeling objectives while augmenting the Llama-2-13b model with an additional 70 million parameters.

**Transformer (Vaswani et al., 2023)** Vaswani's Transformer model revolutionized sequence-to-sequence language modeling tasks by introducing a self-attention mechanism to capture long-range dependencies in data, facilitating more efficient parallelization and exceptional scalability.

**T5-Small (Raffel et al., 2020)** Text-to-Text Transformer (T5) is a language model developed by Google. T5-Small is the smaller variant of T5, comprising approximately 70 million parameters, whereas the base version contains 220M parameters. The T5-Small was created aiming to maintain good performance with a smaller number of parameters.

## 5.3 PERFORMANCE EVALUATION

We evaluated the efficacy of our method using the **compression ratio** (ratio of the original consumed memory to the compressed consumed memory) and the **memory reduction** (difference between the original memory usage and the compressed memory usage). Furthermore, we measured our model's performance in accurately restoring text by evaluating it with BERTScore, BLEU score 6, ROUGE-N, ROUGE-L, and Perplexity (PPL) score.

**BERTScore. (Zhang et al., 2020)** The BERTScore calculates the semantic similarity of two pieces of text by calculating the cosine similarity of their embedding tokens. This metric outputs precision, recall, and f1 score.

**BLEU. (Papineni et al., 2002)** The BLEU metric estimates the quality of candidate text by assigning precision scores to n-grams and comparing them with one or more reference texts. Scores range from 0 and 100, where a higher score denotes better results. The mathematical formula for BLEU is as follows:

$$\text{BLEU} = \text{BP} \times e^{\sum_{n=1}^{N}(w_n \cdot log p_n)} \tag{6}$$

Here, The Brevity Penalty (BP) punishes shorter predictions. $N$ is the maximum n-gram length. $w_n$ are weights for n-gram precision, and $log p_n$ is the logarithm of n-gram precision in the candidate text.

**ROUGE (Lin, 2004)** ROUGE-1 measures unigram overlapping between the generated and reference translations, whereas ROUGE-2 concentrates on bigram overlap. Subsequently, ROUGE-L evaluates the longest common subsequence (LCS), considering word order, unlike ROUGE-1 or ROUGE-2.

**Perplexity (Jelinek, 1976)**

Perplexity is the exponential of cross-entropy loss, reminiscing how uncertain the model is about the test set. A model is more confident in its predictions when has lower perplexity.

## 6 EXPERIMENTAL RESULTS

### 6.1 QUANTITATIVE RESULTS

We scrutinized the outcomes of our TextEconomizer incorporating our meticulous noise process across four corpora. We tested the efficacy of lowering the initial dimensionality into a latent space and further compressing it with general-purpose lossless compressors such as LZMA (Ziv & Lempel, 1978), GZIP (Deutsch, 1996), ZLIB (Deutsch & Gailly, 1996), and ZSTD (Collet & Kucherawy,

Table 1: The comparison of the quantitative performance of various existing methods across different datasets. In this table, **r** symbolizes the memory compression ratio, while **Δ** signifies the total memory saved for each dataset. The mark ∝ denotes identical memory usage across methods, and Θ symbolizes no memory savings.

| Method | #Params. | PwC | | | | WMT19 | | | |
|---|---|---|---|---|---|---|---|---|---|
| | | BLEU | BERT Score | r | Δ | BLEU | BERT Score | r | Δ |
| ICAE | 13.13B | **99.8** | – | 4× | – | – | – | – | – |
| NUGGET | 161M | – | – | – | – | **99** | – | 10× | – |
| Transformer | 86M | 97.33 | **99.46** | ∝ | Θ | 94.13 | 98.86 | ∝ | Θ |
| T5-Small | 70M | 38.29 | 93.58 | ∝ | Θ | 50.15 | 94.55 | ∝ | Θ |
| **TextEconomizer** | **67M** | 95.75 | 99.28 | **67×** | **32GB** | 91.94 | **98.41** | **33×** | **36GB** |

2021). Our rigorous experiment elucidated that LZMA was the most effectual among them in terms of memory utilization, achieving superior compression proportions across all datasets 1. The intuition underpinning these results is that the compression ratio is proportional to the input length, while memory conservation is symmetrical to the magnitude of the dataset.

Table 2: The juxtaposition of the compression ratio of different existing methods.

| Method Name | Corpus Size | BookCorpus |
|---|---|---|
| | | r = (original memory / compressed memory) |
| GPT-AC (Huang et al., 2023) | 7.8M | 10.55 × |
| TRACE (Mao et al., 2022) | 7.8M | 4.49 × |
| TextEconomizer-ZLIB | 1M | 24.62× |
| TextEconomizer-ZGIP | 1M | 24.62× |
| TextEconomizer-ZSTD | 1M | 24.63× |
| **TextEconomizer-LZMA** | 1M | **24.67×** |

TextEconomizer outperforms trending transformer-based models in terms of memory compression ratio and memory conservation. The transformer's intrinsic self-attention mechanism inhibits its ability to narrow the bottleneck layer with enhanced representation. However, TextEconomizer can save 32GB and 36GB of memory per epoch by harnessing the benefit of fixed-size latent representation for the PwC and WMT19 (600K instances) datasets, respectively, thereby surpassing the memory efficiency of the transformer significantly. We also pre-trained a Vaswani-style transformer with a quadruply reduced latent space and residual connection but found it prone to severe overfitting, while information loss is intolerable. Additionally, we experimented with our TextEconomizer incorporating noise injection identical to (Freitag & Roy, 2018), but it yielded no noteworthy outcome worth mentioning. Conversely, we presented the quantitative performance of diverse transformer-based methodologies in Table 1, juxtaposed against TextEconomizer. Our proposed model demonstrates superior performance in compression ratio and memory conservation across all four corpora in small-scale experiments, while maintaining quality with marginal compromise. In particular, it is 196× smaller than the best performing model, with merely a 4% quality performance disparity, illustrating the remarkable efficacy of the parameters. Subsequently, we performed an added performance analysis, focusing on the memory ratio of lossless neural network-based compressors, as depicted in the Table 2. The empirical findings exhibit that TextEconomizer surpasses NN-based lossless compressors in terms of memory ratio efficiency.

We further assessed the performance of T5 small (fine-tuned), Transformer (pre-trained), and TextEconomizer using a less strict metric—the ROUGE score—and observed remarkable results for ROUGE-L, with scores of 98.85, 96.61, 96.37, and 92.86 across the PwC, WMT19, WMT14, and BookCorpus datasets, respectively. The R-L score highlights the intrinsic flexibility of TextEconomizer for the autoencoding task. Our thorough analysis revealed that the Vaswani-style transformer revealed optimal performance, surpassing our model by $6.4 \times 10^{-3}$ in PwC and by 0.0131 in the WMT19 dataset. This indicates a language model with fewer parameters and proper configuration, can minimize quality compromises.

## 6.2 QUALITATIVE RESULTS

The qualitative performance of ICAE, Transformer, T5 Small, and TextEconomizer has been depicted in Table 3, effectively highlighting the excellence of our TextEconomizer and Transformer for the lossy autoencoding task. The examples in the table distinctly demonstrate that T5 Small

Table 3: The qualitative effectiveness of various transformer-based methods in contrast to TextEconomizer. **Red** denotes ignored/ wrong/ extra words / characters, while **Yellow** means lexical items the model did not generate.

| | |
|---|---|
| (Input) | reid and partner alfie hewett came from a set down to beat the french pair stephane houdet and nicolas peifer 4-66-1 7-6(8-6). |
| (ICAE) | reid and partner alfie hewett came from a set down to beat the french pair stephane houdet and nicolas peifer 4-66-1 7-6(8-6). |
| (Transformer) | reid and partner alfie hewett came from a set down to beat the french pair stephane houdet and nicolas peifer 4-66-1 7-6(8-6). |
| (T5 Small) | reid and partner alfie hewett came from a set down to beat the french pair stephan houdet and nicolas peifer 4-66-1 7-6(8-6). |
| (TextEconomizer) | reid and partner alfie hewett came from a set down to beat the french pair stephane houdet and nicolas peifer 4-66 -6-8 (8-6) |
| (Input) | experimentally, we comprehensively compare the behavior of icl and explicit fine-tuning based on real tasks to provide empirical evidence that supports our understanding. the results prove that icl behaves similarly to explicit fine-tuning at the prediction level, the representation level, and the attention behavior level. |
| (ICAE) | experimentally, we comprehensively compare the behavior of icl and explicit finetuning based on real tasks to provide empirical evidence that supports our findings . the experimental evidence proves that icl behaves like us to the same extent . prediction at the explicit finetuning level, the representation level, and the attention behavior level. |
| (Transformer) | experimentally, we comprehensively compare the behavior of icl and explicit fine-tuning based on real tasks to provide empirical evidence that supports our understanding. the results prove that icl behaves similarly to explicit fine-tuning at the prediction level, the representation level, and the attention behavior level. |
| (T5 Small) | experimentally, we comprehensively compare the behavior of icl and explicit fine-tuning based on real tasks to provide empirical evidence that supports our understanding. the results prove that icl behaves similarly to explicit fine-tuning at the prediction level, the representation level, and the attention behavior level. |
| (TextEconomizer) | experimentally, we comprehensively compare the behavior of icl and explicit fine-tuning based on real tasks to provide empirical evidence that supports our understanding. the results prove that icl behaves similarly to explicit fine-tuning at the prediction level, the representation level, and the attention behavior level. |
| (Input) | sarah found a $50 bill on the street and excitedly shouted, "i'm going to save this!" ten minutes later, she walked out of the store with $75 worth of things she didn't need, proudly calling it an "investment." |
| (Transformer) | sarah found a $50 bill on the street and excitedly shouted, "i'm going to save this ? " ten minutes later . she walked out of the store with $75 worth of things she didn't need, proudly calling it an "investment." |
| (T5 Small) | sarah found a $50 bill on the street and excitedly shouted , " i'm going to save this!" ten minutes later, she walked out of the store with $75 worth of things she didn't need, proudly calling it an "investment." |
| (TextEconomizer) | sarah found a $50 bill on the street and excitedly shouted, ' i'm going to save this! ' ten minutes later, she walked out of the store with $75 worth of things she didn't need, proudly calling it an investment investment. |
| (Input) | tiny toes and button nose, a bundle of joy soon to expose! |
| (Transformer) | tiny toes and button nose : a bundle of joy soon to expose ? |
| (T5 Small) | tiny toes and button nose , a bundle of joy soon to expose! |
| (TextEconomizer) | tiny toes and button nose, a bundle of joy soon to be seen . |
| (Input) | in some cases the number is 120,000,130,000. |
| (Transformer) | in some cases the number is 120,000,130,000. |
| (T5 Small) | in some cases the number is 120,000,130,000. |
| (TextEconomizer) | in some cases the number is 120,000,130,000. |

struggles with punctuation and long sentences, while ICAE often replaces words with synonyms, occasionally generating multiple extra words, which are semantically uniform yet longer sequences. However, this added length can raise redundancy in various scenarios. Subsequently, Transformer and TextEconomizer demonstrate the capacity to produce sentences identical to the input when concise and extensive scenarios. However, Transformer occasionally misplaces punctuation, changing the tone at sentence endings (e.g., $4^{th}$ example). TextEconomizer also exhibits minimal punctuation challenges, particularly at the sentence conclusion, but it avoids modifying meaning or producing inconsistent expressions. One notable observation, as seen in the third example, is that TextEconomizer tends to generate single quotes in place of double quotes and sometimes repeats the same word twice (e.g., ... investment investment...") within the double-quoted text, while still keeping the original semantic content—the major concern in lossy evaluation. Our TextEconomizer achieves this feat admirably, utilizing significantly fewer parameters. It is noteworthy that all methods, including T5 Small, exhibit proportional efficacy when processing shorter sentences, as demonstrated by the last example.

## 6.3 ABLATION STUDY

Table 4 elucidated how model performance improves with larger corpus sizes. In this extensive

Table 4: The influence of corpus size PwC on the performance of our proposed method.

| Method | Corpus Size | Inference | | | |
|---|---|---|---|---|---|
| | | **BLEU** | **BERT Score** | **R-L** | **PPL** |
| TextEconomizer | 100K | 87.30 | 97.27 | 93.69 | 22.44 |
| TextEconomizer | 200K | 91.97 | 98.41 | 96.59 | 17.92 |
| TextEconomizer | 242K | 95.75 | 99.27 | 98.85 | 11.26 |
| TextEconomizer (No Attention) | 242K (PwC) | 7.87 | 79.78 | 19.29 | 948.85 |
| TextEconomizer (No Attention) | 600K (WMT19) | 31.16 | 85.68 | 46.72 | 240.02 |

study, we used four large-scale datasets, with a particular focus on the PwC dataset to illustrate the correlation between corpus size and model effectiveness. Interestingly, the corpus with 242K instances outperformed those with 100K and 200K instances. Contrarily, the corpus with 100K instances showed the least significant results, while the 200K-instance corpus produced moderate outcomes. This tendency was consistently observed in all four datasets, signifying that larger corpus sizes guide enhanced performance (Bijoy et al., 2023). Additionally, we conducted another ablation study on TextEconomizer's performance without the attention mechanism across the datasets and observed suboptimal performance despite a marginal improvement in training time. These findings spotlight that the attention mechanism is the pivotal component of our model, pushing incomparable performance across all datasets with a marginal trade-off.

## 7 CONCLUSION

This study presents a memory-efficient baseline for the task at hand, proposing the TextEconomizer, a monolingual autoencoder-based approach that leverages attention mechanisms and a novel text noising strategy. TextEconomizer refines the fixed-size latent representation and additionally leverages the compatibility of entropy coding algorithms to condense the latent space more efficiently, therefore adeptly maneuvering the intricate linguistic complexities inherent to the task. It surpassed transformer-based methods in parameter and memory efficiency across various corpora, with only a negligible quality trade-off. Notably, we demonstrated that pre-training traditional transformers with minimal settings can acquire performance $\approx 2\%$ below of best-performing models in autoencoding tasks—by integrating our sophisticated noisy text processing, therefore questioning the notion that autoencoder-based approaches are only adequate for image compression, extending their relevance to text-based tasks. Our work opens further avenues for efficient natural language processing in resource-constrained settings. Our future research directions include knowledge distillation from multilingual to our monolingual model and large-scale experiments integrating contrastive learning techniques.

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

## A APPENDIX

You may include other additional sections here.

