# OpenReview forum: "TextEconomizer: Enhancing Lossy Text Compression with Denoising Autoencoder and Entropy Coding"
_ICLR.cc/2025/Conference — ICLR 2025 Conference Withdrawn Submission_

### Official Review · Reviewer_k4jx · 2024-11-03

**Soundness:** 2
**Presentation:** 2
**Contribution:** 2
**Rating:** 3
**Confidence:** 4

**Summary:**

This paper presents TextEconomizer, a lossy text compression method that combines a denoising autoencoder with entropy coding. The main claimed contributions are:
- A text noise process for training robust denoising
- A monolingual autoencoder architecture using fixed-size latent representations
- Integration of entropy coding for improved compression ratios
- Evaluation showing 67x compression while maintaining quality metrics
- Analysis of training corpus size effects on performance

**Strengths:**

- The reported performance is good.

**Weaknesses:**

- **Lack of technical contribution**: The reviewer didn't see any technical contribution of the proposed method.
- **Lack of analysis and inspiration**: The authors didn't provide any principles, analysis, theory, or even intuitive explanations for their proposed methods. Readers are unable to understand why the proposed method outperforms others.
- **Duplicate claims of contribution**: Despite the claimed 5 contributions, most of them are duplicate and meaningless.
- **Bad presentation**: The proposed method is not presented well in the context of the paper. The authors prioritized massive details of datasets over task formulation and methodology, which makes it difficult for readers to understand the technical contribution of the paper. **LZMA** shows up in Figure 1 without any prior definition. Redundancy in this paper somehow validates the importance of text representation compression.

**Questions:**

See weaknesses.

---

> ### Author Response · Authors · 2024-11-19
> **Noise Robustness and Lower Complexity**
>
> Thank you for your valuable feedbacks. TextEconomizer is well-suited for auto-encoding tasks due to its bidirectional encoder, which captures contextual information from both past and future tokens, and its attention mechanism, which enables the decoder to focus on the most relevant parts of the input sequence. The model effectively encodes the input into a compact latent representation, leveraging GRU layers to manage sequential dependencies. During decoding, the attention mechanism ensures that the output is aligned with the input, preserving its structural integrity while reconstructing the sequence. This combination allows the model to excel in reconstructing input sequences with high fidelity, even when faced with noise or variability in the data. As we have annotated all the datasets with rigorous noise injection, the model learned to convert real-world noise into properly denoised text, exhibiting comprehensive performance in memory consumption with negligible performance trade-offs.
>
> We have observed significant advancements in image compression using variational auto encoders. In contrast, our approach to text compression uniquely incorporates real-world noise, demonstrating competitive results across all relevant metrics while maintaining lower architectural complexity. This is particularly noteworthy as transformer-based networks are more complex and require large amounts of data.

---

### Official Review · Reviewer_z5S6 · 2024-11-04

**Soundness:** 2
**Presentation:** 2
**Contribution:** 2
**Rating:** 3
**Confidence:** 2

**Summary:**

The paper argues that while lossy compression is widely used in image processing, its application in text compression has been less explored. Their proposed model, TextEconomizer, compresses variable-sized text inputs into a fixed-size latent space via an autoencoder-style framework and achieves relatively good compression ratios with competitive text quality, as measured by BLEU, BERT Score, ROUGE scores and PPL. The model is parameter-efficient, and it significantly outperforms existing transformer-based models in memory efficiency with the use of entropy coding on latent representations.

**Strengths:**

High Compression Ratio: TextEconomizer achieves a remarkable 67× compression ratio (lossy) while maintaining relatively high BLEU and ROUGE scores, indicating that it is effective at compressing text without significant loss of meaning.

Memory Efficiency: The model demonstrates significant memory efficiency with additional entropy coding on latent representations.

Parameter Efficiency: TextEconomizer shows that high performance can be achieved with a parameter count significantly smaller than comparable models.

**Weaknesses:**

(1) Using autoencoder and entropy coding for lossy compression is not a new idea, especially for visual signal compression like image compression [Ref1] and video compression [Ref2]. When referring to lossy image compression methods with variational autoencoders, the authors should include these representative related works in this paper (incomprehensive literature review).

[Ref1] Variational Image Compression with a Scale Hyperprior. Ballé et al., ICLR 2018.

[Ref2] DVC: An End-to-end Deep Video Compression Framework. Lu et al., CVPR2019.

(2) When applying lossy compression for text compression, it is obvious that text corpus get much high compression ratio compared with lossless compression methods, but with the sacrifice of text reconstruction precision. Different from visual signals likes images, text is a data modality with high information density. Therefore, if the authors would like to compression text in a lossy way, they should convince others that " Lossy text compression reduces data size while preserving core meaning, making ideal for some tasks". In other words, some experiments should be performed on tasks like summarization, automated analysis, and digital archives, to ensure that lossy text compression is still useful for these tasks.

(3) Different from lossless compression, lossy compression should usually be measured by different compression ratios and different distortion levels, more like a compression ratio - distortion curve. Maybe the authors can adjust the latent dimension to investigate on different compression ratios.

(4) Most of the literature referred in this paper is not correctly cited. Many references are Arxiv version. In addition, the well-known paper "Attention is All you Need" is mistaken as a paper published in 2023 (which should be a paper published in NeurIPS 2017)

**Questions:**

As I am not very familiar to this area, I hope the abovementioned weakness points could get some feedback from the authors.

**Details Of Ethics Concerns:**

No Ethics Concerns

---

> ### Author Response · Authors · 2024-11-19
> **Utility and Semantic Integrity of Lossy Text Compression**
>
> Thank you for your insightful comment. Our goal with TextEconomizer is not only to achieve higher compression ratios but also to ensure that the compressed representation retains its semantic essence. This is clear in our evaluation metrics, where we evaluated our model with BLEU, BERT, and ROUGE 1-2-L scores. These metrics emphasize semantic similarity and word-for-word alignment. Our approach reflects how well-reconstructed lossy text can still convey meaning—a critical factor for applications like summarization, search indexing, and digital archiving. TextEconomizer with a negligible word-to-word distortion while still conveying the same meaning provides a significant advantage by utilizing less memory during training and saving relevant storage space. We look forward to strengthening our methods through more targeted task evaluations.

---

### Official Review · Reviewer_XGgy · 2024-11-06

**Soundness:** 2
**Presentation:** 1
**Contribution:** 2
**Rating:** 3
**Confidence:** 4

**Summary:**

This paper proposes an auto encoder for text data, with the goal of lossy text compression. The goal is to preserve text semantics while minimizing rate transmitted. The method, called TextEconomizer, consumes a noisy text, transmits a fixed size latent vector, and reconstructs the text, using cross-entropy loss with partial teacher-forcing. Experiments compare TextEconomizer to several baselines, primarily lossless text compressors, and some text language models.

**Strengths:**

- Lossy text compression seems to be a relatively novel area of research, so the novelty of such a work is good.

**Weaknesses:**

- I found the overall presentation of the work to be a bit confusing. Section 3 seems to include a lot of data creation details, which appears to explain the noise adding process, but a lot of the details I felt should have been placed in experimental setup or appendix.
- Many of the baselines mentioned in the related work are not compared to. The only relevant baseline used in section 6 that actually does lossy text compression appears to be NUGGET, although I may be mistaken. Everything else appears to be a language model (such as T5) or lossless text compressor (Huang et al, 2023). It is difficult to judge the efficacy of TextEconomizer without a comparison to the lossy text compressors in the related work.
- Furthermore, the baselines section of 5.2 does not include all the baseline comparisons actually used in section 6.
- In Table 1, it is hard to say that TextEconomizer is superior to NUGGET (the only other lossy text compressor). NUGGET has less memory compression ratio but superior BLEU.
- In addition, NUGGET is missing in Table 3. It would be helpful to have a qualitative comparison for NUGGET.
- In my opinion, it may also be useful to have a metric such as Levenshtein score, in order to measure the similarity between texts in text space, rather than just BERTScore, which compares in the embedding space. This comparison would help support the qualitative results in Table 3.
- An ablation study is missing. I think this is important because the texts shown seem to be relatively short. Since the latent variable is fixed-size, it is possible the performance may suffer if the input text lengths are longer. It would be helpful to know how the model performance changes based on (i) the fixed-size latent variable size and (ii) the input text length. In addition, the ablation study could support other design choices, such as the noise adding process.

**Questions:**

In addition to the questions in the weaknesses section, I have the following questions:
- Why is the fixed-size latent space necessary? I think this was motivated in the introduction, but I found the explanation there to be confusing. Presumably, a variable-size latent space could still be entropy coded or LZ-coded.
- Moreover, is the latent Z quantized before encoded with LZ? If not, this does not seem possible, as lossless compression needs a discrete input.
- Why not additionally use an entropy-model to minimize the entropy of the latent z? This would further reduce the rate, and jointly minimize the rate and cross-entropy. It can also support a variable-size latent space.

---

> ### Author Response · Authors · 2024-11-19
> **Responses on TextEconomizer's Design and Efficiency**
>
> In NUGGET the authors chose BART ref[1], a transformer-based sequence-to-sequence model as the foundational architecture for their experiments on top of 602M parameters checkpoint ref[2]. TextEconomizer takes a more lightweight approach, using a simpler bi-directional GRU-based architecture with just ~67M parameters. Despite this, TextEconomizer achieves a strong balance, with only a 7.06% drop in BLEU score while maintaining a promising BERT Score. Since we're working on lossy text compression, semantic similarity (captured by BERT Score) is equally important, as BLEU's strict word-by-word matching might not fully reflect text quality in this context. We'll make sure to highlight this trade-off more clearly.
>
> ref[1]: Lewis, M., Liu, Y., Goyal, N., Ghazvininejad, M., Mohamed, A., Levy, O., Stoyanov, V., and Zettlemoyer, L. BART: Denoising Sequence-to-Sequence Pre-training for Natural Language Generation, Translation, and Comprehension. In Annual Meeting of the Association for Computational Linguistics (ACL), 2020.
>
> ref[2]: Tang, Y., Tran, C., Li, X., Chen, P.-J., Goyal, N., Chaudhary, V., Gu, J., and Fan, A. Multilingual Translation with Extensible Multilingual Pretraining and Finetuning, 2020.
>
>
> Reply [Q1]: The Transformer produces a tensor with the shape [max_length (can vary in each batch), batch_size, hidden_dim] at the encoder end before passing the contextualized tensor to the decoder for each batch. In contrast, TextEconomizer passes a tensor with the shape [batch_size, hidden_dim] to the decoder. This means that while processing each batch, TextEconomizer can generate high-quality contextualized tensors while eliminating [max_length * 4] bytes. Consequently, storing our model's contextualized tensors requires less memory than the Transformer during training, and also if we want to save the bottleneck tensors explicitly. Additionally, entropy coding provides an extra advantage in saving even more memory when saving explicitly.
>
> Reply [Q2]: No the latent Z was not quantized before encoded with LZMA.
>
> Reply [Q3]: Thank you for sharing your intuition. With our TextEconomizer, we have measured the bits-per-character (bpc) as follows: 0.8988 for WMT19, 0.3350 for WMT14, 0.7016 for PwC, and 0.8670 for the BookCorpus dataset. If you don't mind, could you share a little more briefly about incorporating the entropy model with TextEconomizer? That information would be helpful for our quick experiments.

---

> > ### Comment · Reviewer_XGgy · 2024-11-25
> >
> > Thanks for the response. Regarding Q2 and Q3, they go hand in hand, and perhaps that is causing me some confusion. Given your reply to Q1, my understanding is that for each text, the Z variable is a vector of dimension hidden_dim. Is Z floating-point? If so, how is LZMA applied to this vector? Eq (3) says tanh() is applied, so the values of Z are continuous between 0 and 1. As a lossless compressor, LZ can only be applied to discrete inputs, which is why I don't understand how it can be applied without quantizing Z first.
> >
> > In the neural compression literature, the latent variable is typically quantized, and then a likelihood model is fitted over the quantized symbols which are discrete. During training, the likelihood model is optimized to minimize the entropy of the quantized symbols. This helps facilitate a joint rate-distortion trade-off during training. Regarding Q3, it seems to get best performance, one should do a similar approach here. But given that Z is not quantized, I have some significant confusion as to how LZ (or other entropy coding) is applied, and how the bitrates in the paper are reported.

---

> > > ### Author Response · Authors · 2024-11-26
> > >
> > > Thanks for your valuable comment.
> > >
> > > We compressed the latent space by first converting the floating-point latent vector \( Z \) into its binary representation. To achieve this, we detached \( Z \) from the computation graph and moved it to the CPU to ensure compatibility with NumPy, which we used to serialize each element of \( Z \) into its raw byte format. Then we applied the LZMA compressor to generate the compressed representation of \( Z \), and we computed the compressed memory consumption at this stage. After this calculation, we used the LZMA decompressor to reconstruct the original binary representation and subsequently converted it back into the original latent vector \( Z \) with its original shape, data type, and gradient properties preserved. This reverse process ensures the recoverability of the latent space before feeding to the decoder. We calculated the ratio, $r$, by dividing the *input memory consumption* by the *compressed memory consumption*.
> > >
> > > When calculating the bits per character (bpc), we used the formula:
> > >
> > > `bpc = (test_loss / ln(2)) ÷ avg_num_of_chars_per_token`
> > >
> > > *ln(2)* is used to convert the test loss from nats (natural log base) to bits (log base 2). Where it aligns the loss with bits per character, reflecting the average information content per character. It is further normalized by the average number of characters per token.
> > >
> > > `Average Characters per Token =  Total Number of Characters in Text / Total Number of Tokens (WordPiece)`
> > >
> > > We hope this explanation addresses your concerns. If you have any other questions, we would be happy to answer them.

---

### Official Review · Reviewer_KeMW · 2024-11-08

**Soundness:** 3
**Presentation:** 1
**Contribution:** 3
**Rating:** 3
**Confidence:** 5

**Summary:**

The paper proposed a DAE method designed for English text self-coding task is constructed by one-way bidirectional gated recursive unit (GRU) and combined with entropy coding to optimize the compression effect.

**Strengths:**

S1. The memory compression ratio achieves on texts are impressive.

**Weaknesses:**

W1. [Clarity & Writing - 1] The paper's writing could be improved to clearly outline its contributions in the introduction and abstract. Additionally, the text becomes overly verbose and includes several claims in the method section without direct references. Proper citations are needed for every claim that refers to existing literature. For instance, there is a missing reference or explanation for the "Lempel–Ziv–Markov compressor."

W2. [Clarity & Writing - 2] The paper's structure should be clearer and better organized. For example, Section 3 dedicates substantial space to details about the dataset (e.g., number of words), which is less relevant to the content of Section 4. Furthermore, the dataset attributes are repeated in Section 5.1. Additionally, Section 5 appears to be a continuation of the experimental results, and it should not be clearly separated from Section 6. Additionally, the tables should be properly positioned and the fonts should be consistent.

W3. [Experiments] There are a lot of unfair comparisons and over claims in the paper. For example, the Transformer shown in Table 1 outperforms the proposed method (97.33 vs 95.75, 99.46 vs 99.28) in terms of BLEU and BERT Score. Considering that the number of parameters of the transformer can be adjusted by reducing the number of layers or the dimension of embeddings, it is more fair to choose the transformer structure with the same number of parameters as the method in this paper.

**Questions:**

1. Teacher-forcing is mentioned in Figure 1, but is not described in the method section. Can you describe it briefly?

---

> ### Author Response · Authors · 2024-11-19
> **Clarification on Teacher-forcing and Experimental Results with Parameter Matching in Transformer**
>
> Reply [Q1]: In our model, teacher forcing refers to the process where, during training, the target sequence tokens are directly used 50% of the time as input to the decoder at each time step instead of the decoder’s own predicted output and 50% of the time decoder has got it's own predicted output as input. Incorporating, this approach helps guide the decoder more effectively towards generating accurate outputs and stabilizes the training process.
>
>
> Reply: W3. [Experiments] Thank you for your insightful suggestion regarding running experiments with transformer parameters same as the TextEconomizer. We conducted experiments using Vaswani's style transformer by reducing the feed-forward layer dimensions and the number of encoder-decoder layers to 5 instead of 6. Interestingly, we observed that the results remained identical on PwC dataset. I have noted the results below:
>
> | Model       | #of Parameter | PPL  | BLEU  | ROUGE-L | BERT SCORE |
> |------------|---------------|------|-------|----------|------------|
> | Transformer | ~67M         | 4.2  | 97.43 | 0.9954  | 0.9948     |

---

> ### Author Response · Authors · 2024-11-19
> **Additional Experimental Results.**
>
> We’ve run experiments where 20% and 50% of tokens were passed to the decoder in a Vaswani-style Transformer. The token selection was done using a softmax layer based on a probability distribution. Additionally, we modified the architecture by replacing sinusoidal positional embeddings with Rotary Positional Embeddings, ReLU with SwiGLU, and LayerNorm with RMSNorm (we’ve been calling this version LLaMAFormer in the below table) and depicted the results below. We also tested the 20% and 50% token selection on this LLaMAFormer architecture. All these experiments were run on the PWC (full) and WMT19 (600K) datasets, and the results have been illustrated in the below table.
>
> | Model Name   | Dataset | #of Token | BLEU Score | BERT Score | ROUGE-L | PPL   |
> |-------------|---------|-----------|------------|------------|---------|-------|
> | Transformer | PwC     | 20%       | 95.6077    | 0.9911     | 0.9835  | 4.846 |
> | LLaMAFormer | PwC     | 20%       | 92.6727        | 0.9855        | 0.9649     | 4.896   |
> | Transformer | WMT19   | 20%       | 90.9721    | 0.9822     | 0.9597  | 6.01  |
> | LLaMAFormer | WMT19   | 20%       | 93.1267    | 0.9868     | 0.9712  | 5.036 |
>
> | Model Name   | Dataset | #of Token | BLEU Score | BERT Score | ROUGE-L | PPL   |
> |-------------|---------|-----------|------------|------------|---------|-------|
> | Transformer | PwC     | 50%       | 96.9844    | 0.9941     | 0.9926  | 4.34  |
> | LLaMAFormer | PwC     | 50%       | 95.9452    | 0.9922     | 0.9861  | 4.215 |
> | Transformer | WMT19   | 50%       | 93.1854    | 0.9867     | 0.9747  | 5.41  |
> | LLaMAFormer | WMT19   | 50%       | 93.6126    | 0.9878     | 0.9744  | 4.879 |
>
> | Model Name   | Dataset | #of Token | BLEU Score | BERT Score | ROUGE-L | PPL   |
> |-------------|---------|-----------|------------|------------|---------|-------|
> | LLaMAFormer | PwC     | 100%      | 97.1551    | 0.9943     | 0.9941  | 4.185 |
> | LLaMAFormer | WMT19   | 100%      | 93.9515    | 0.9883     | 0.9764  | 4.822 |

---

### Author Response · Authors · 2024-12-01

We appreciate your response, which has helped us further improve our work. Your time and contribution means a lot for us. At the same time, if our explanation has addressed your concerns, we kindly hope that you would consider increasing the score or confidence of our work. If you still have any questions regarding our work, please feel free to contact us, and we will respond as soon as possible.

Thanks again for your assistance here.

---

### Note · Authors · 2025-01-23

I have read and agree with the venue's withdrawal policy on behalf of myself and my co-authors.